# Improved Training of Deep Text Clustering

**Zonghao Yang    Wenpeng Hu**[*]   **Yushan Tan    Zhunchen Luo**

Information Research Center of Military Science, PLA Academy of Military Science
{yangzonghao1024, tanys0106}@163.com
wenpeng.hu@pku.edu.cn, zhunchenluo@gmail.com

## Abstract

The classical deep clustering optimization methods basically leverage information such as clustering centers, mutual information, and distance metrics to construct implicit generalized labels to establish information feedback (weak supervision) and thus optimize the deep model. However, the resulting generalized labels have different degrees of errors in the whole clustering process due to the limitation of clustering accuracy, which greatly interferes with the clustering process. To this end, this paper proposes a general deep clustering optimization method from the perspective of empirical risk minimization, using the correlation relationship between the samples. Experiments on two classical deep clustering methods demonstrate the necessity and effectiveness of the method. Code is available at https://github.com/yangzonghao1024/DCGLU.

## 1 Introduction

Cluster analysis plays a role in machine learning and data mining as it can classify data into different groups in an unsupervised way. Over the past decades, a large number of clustering methods with shallow models have been proposed (Ren et al., 2019; Comaniciu and Meer, 2002; Ren et al., 2018; Bishop and Nasrabadi, 2006; Ren et al., 2017; Cai et al., 2013; Huang et al., 2021). The performance of them on the complex data is limited due to the poor power of feature learning.

Recently, deep learning based clustering approach (referred to deep clustering) aims at effectively extracting more clustering-friendly features from data and performing clustering with learned features simultaneously (Chang et al., 2017; Albert et al., 2022; Ronen et al., 2022).

After in-depth research, we find that most deep clustering models are usually optimized through specific supervision information. We call it generalized supervision in clustering which is implemented through generalized labels in most cases. The generalized labels usually have different generation method according to different deep clustering algorithm. We will elaborate them in Section 2. Here, we argue that the generalized labels are full of noise due to the performance limitation of clustering model and estimation methods especially at the beginning of clustering progress, which will disturb the recognition of clustering group and influence the clustering result. To this end, inspired by Positive and unlabeled Learning (PU learning), we propose a novel method to reduce the impact of noise in generalized labels, namely Deep Clustering optimization from Generalized Labeled and Unlabeled data (DCGLU for short).

The proposed DCGLU divides the clustering samples into high confidence samples and low confidence samples according to the confidence level of the samples been correctly clustered. Obviously, the generalized labels of low confidence samples tend to have more errors (as the decision confidence usually related to accuracy (Hu et al., 2020)). In this case, we propose to regard the low confidence samples as unlabeled samples and the high confidence samples as labeled samples, then learning the model in the same fashion with PU learning to reduce the errors in generalized supervision and thus improve the clustering performance.

Overall, our contributions can be summarized as follows:(1) it proposes a new problem that exists in most deep clustering approaches and formulate it as a problem of generalized supervision with generalized label, which will facilitate the research of deep clustering; (2) it proposes a general deep clustering optimization method, which can be leveraged to reduce the impact of noise in clustering; (3) experiments show the proposed DCGLU can improve the clustering features and clustering results of strong baselines.

---

[*] Corresponding authors.

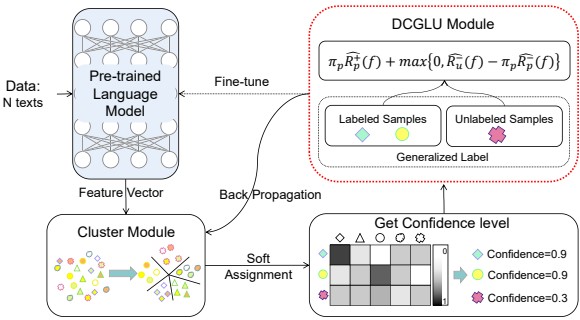

Figure 1: Architecture overview of DCGLU.

## 2 Problem Formulation

Given the unlabeled text dataset $D = \{x_i, i = 1, ..., N\}$, where $x_i$ represents the $i^{th}$ text in the corpus, our goal is to cluster $N$ samples into $k$ different classes by an unsupervised method. Each sample in $D$ is represented by a feature vector. We use the pre-trained language model Bert (Devlin et al., 2018) to obtain the representation $e_i = \text{BERT}(x_i)$ of each text.

As discussed in Section 1, the deep model in deep clustering methods is usually optimized by specific supervised information. We refer the specific supervision to **generalized supervision**, which is basically realized by generalized labels. *The generalized label generation method varies among different clustering methods* [1]: 1) The cluster center used in K-means-based methods (Yang et al., 2017; Xie et al., 2016) can be considered as the generalized label; 2) Spectral clustering correlation methods (Chang et al., 2017; Yang et al., 2019) use the distance similarity matrix between samples to minimize the similarity distance between samples, the similarity between samples can be regarded as generalized labels; 3) Gaussian Mixture Model methods (Ronen et al., 2022) assume that the samples in each class obey an independent Gaussian distribution. The center of the Gaussian distribution can be regard as the generalized label; 4) (Caron et al., 2018) use the clustering assignment results as pseudo-label information to optimize feature generation. Thus clustering assignment results are the generalized labels.

Formally, deep clustering algorithm will generate a target distribution $P$, which is used to indicate the probability distribution of samples of the same category, and optimize the clustering model through stochastic gradient descent algo-

rithm. Hence, the generated target distribution $P$ can be considered as the distribution of the generalized labels $Y'$ of the samples. The process of fine-tuning the deep clustering model is the process of generalized label exploitation, which can be described formally by the experience risk $R_\delta(f)$:

$$R_\delta(f) = \frac{1}{n} \sum_{i=1}^{n} \mathcal{L}\left(f\left(x_i\right), y_i'\right) \qquad (1)$$

where $\mathcal{L}$ is the distance metric function of $f(x)$ and the generalized label $y'$, $(x_i, y_i') \sim P$.

## 3 Method

The noises in generalized label $Y'$ result in error propagation problem. A common used method for the problem is instance weighting (Lison and Bibauw, 2017): $R_\delta'(f) = \frac{1}{n} \sum_{i=1}^{n} w_i \cdot \mathcal{L}\left(f\left(x_i\right), y_i'\right)$. $w_i$ is the importance estimation weight of each sample. Clearly, there are two main challenges for instance weighting, (1) it reduces the impact of noise while ignoring a large amount of information in data with small $w_i$; (2) the estimation of $w_i$ is very tricky due to the accuracy limitation of unsupervised clustering.

For challenge (2), we propose to use the confidence of samples been correctly clustered (easy estimation, see appendix A.1) to measure the quality of generalized labels. Note that the confidence cannot replace noise estimation as the generalized labels tend to have little errors with high confidence but the contrary is not necessarily true in most cases. To solve the above problem and challenge (1), we regard the samples with low confidence as the unlabeled data as we cannot make sure the correctness of their generalized labels.[2] Therefore, the samples are divided into high-confidence labeled samples and unlabeled (low confidence) samples according to their confidence levels by hyper-parameter $t$. [3] Then we adapt PU learning (Liu et al., 2002) to our case for model optimization. Here we directly provide the final adaptation formula, for detailed

---

[1]please note that the generalized label is not necessarily a discrete classification label, but also a continuous measurement, such as distance

[2]According to the maximum entropy principle, the optimal treatment for samples with unknown distribution is to make no assumptions, in which case the probability distribution is the most uniform and the risk of prediction is minimal.

[3]We set $t$ to 0.8 in the experiments which performs well in different data sets and for different models.

| Dataset | SNIPS | | | DBPedia | | |
|---|---|---|---|---|---|---|
| Method | NMI | ARI | ACC | NMI | ARI | ACC |
| KM | 71.42 | 67.62 | 84.36 | 67.26 | 49.93 | 61.00 |
| AC | 71.03 | 58.52 | 75.54 | 65.63 | 43.92 | 56.07 |
| SAE-KM | 78.24 | 74.66 | 87.88 | 59.70 | 31.72 | 50.29 |
| DEC | 84.62 | 82.32 | 91.59 | 53.36 | 29.43 | 39.60 |
| DCN | 58.64 | 42.81 | 57.45 | 54.54 | 32.31 | 47.48 |
| BERT-KM | 52.11 | 43.73 | 70.29 | 60.87 | 26.6 | 36.14 |
| DAC | $75.51 \pm 4.93$ | $67.13 \pm 7.10$ | $77.17 \pm 6.61$ | $72.73 \pm 3.57$ | $53.01 \pm 5.04$ | $63.27 \pm 5.74$ |
| DAC* | $\mathbf{79.92} \pm 4.40$ | $\mathbf{70.67} \pm 6.89$ | $\mathbf{79.64} \pm 6.22$ | $\mathbf{74.25} \pm 2.57$ | $\mathbf{56.93} \pm 4.36$ | $\mathbf{66.90} \pm 4.96$ |
| DeepDPM | $83.69 \pm 2.62$ | $79.48 \pm 5.80$ | $88.35 \pm 6.10$ | $69.17 \pm 4.33$ | $48.02 \pm 6.68$ | $58.30 \pm 6.52$ |
| DeepDPM* | $\mathbf{85.15} \pm 0.10$ | $\mathbf{82.93} \pm 0.12$ | $\mathbf{92.07} \pm 0.05$ | $\mathbf{72.14} \pm 2.11$ | $\mathbf{52.84} \pm 3.21$ | $\mathbf{62.36} \pm 4.54$ |

Table 1: The clustering results on two datasets, where with * denotes the result after introducing DCGLU. SNIPS and DBPedia baseline results are obtained from the unsupervised section of (Lin et al., 2020). All reported results are in percentages. DAC as well as DeepDPM results are averaged from the source code or third-party quality code under 10 runs ($\pm$std.dev.).

derivation process, please refer to Appendix A.2:

$$
\begin{aligned}
\widehat{R}_{\mathrm{pu}}(f) &= \pi_{\mathrm{p}}\widehat{R}_{\mathrm{p}}^+(f) + \max\left\{0, \widehat{R}_{\mathrm{u}}^-(f) - \pi_{\mathrm{p}}\widehat{R}_{\mathrm{p}}^-(f)\right\} \\
\widehat{R}_{\mathrm{p}}^+(f) &= \frac{1}{n_{\mathrm{p}}}\sum_{i=1}^{n_{\mathrm{p}}} \mathcal{L}\left(f\left(x_i^{\mathrm{p}}\right), y_+^{\prime\mathrm{p}}\right) \\
\widehat{R}_{\mathrm{p}}^-(f) &= \frac{1}{n_{\mathrm{p}}}\sum_{i=1}^{n_{\mathrm{p}}} \mathcal{L}\left(f\left(x_i^{\mathrm{p}}\right), y_-^{\prime\mathrm{p}}\right) \\
\widehat{R}_{\mathrm{u}}^-(f) &= \frac{1}{n_{\mathrm{u}}}\sum_{i=1}^{n_{\mathrm{u}}} \mathcal{L}\left(\bar{f}\left(x_i^{\mathrm{u}}\right), -1\right)
\end{aligned}
\tag{2}
$$

where $y_+^{\prime\mathrm{p}}$ is the multi-class generalized label corresponding to the high confidence sample; in general, $y_-^{\prime\mathrm{p}}$ can be expressed as $y_-^{\prime\mathrm{p}} = 1 - y_+^{\prime\mathrm{p}}$; $\pi_{\mathrm{p}}$ is the prior probability of positive samples.

Overall, we give the clustering optimization algorithm in the Algorithm 1 in appendix A.2, and give the architecture of DCGLU in Figure 1. Clearly, DCGLU realizes decoupling from the clustering model, therefore it can be applied to most clustering algorithms.

# 4 Experiments

## 4.1 Experiment Setup

We apply the DCGLU method on two strong deep clustering method: DAC (Chang et al., 2017) and DeepDPM (Ronen et al., 2022), and then evaluate them on two publicly available text datasets: SNIPS (Coucke et al., 2018)(7 classes) and DBPedia (Lin et al., 2020) (14 classes). We further take K-Means(KM) (MacQueen, 1967) and agglomerative clustering(AG) (Gowda and Krishna, 1978), SAE-KM, BERT-KM, Deep embedding clustering(DEC) (Xie et al., 2016), Deep clustering network(DCN) (Yang et al., 2017), etc as the baselines into comparison to show the competitiveness

of DCGLU. We keep our primary experimental setup consistent with what DeepDPM[4] and DAC[5] reported in their original paper. For more details of datasets, baselines and implement settings please refer to Appendix B.

## 4.2 Compared with strong baselines

Table 1 shows the clustering results of strong baselines and the effectiveness of DCGLU. We can draw the following observations.

Firstly, the proposed DCGLU has significant improvement on two different kinds of deep clustering methods DAC and DeepDPM (with p-value < 0.01 on paired t-test). The consistency improvement of multiple algorithms, evaluation metrics and datasets indicates that DCGLU has better effectiveness and universality.

Secondly, we can see that 1) on the SNPIS dataset, with the help of DCGLU, DeepDPM succeeds in significantly outperforming the strongest baseline system (DEC) participating in the comparison (with p-value < 0.01 on paired t-test); 2) on the DBPedia dataset, DCGLU makes DAC as well as DeepDPM, the top two powerful systems, get better results. This indicates that the error propagation problem in generalized supervision described in this paper has a large impact on the performance of deep models and is prevalent (even in very strong clustering systems), while DCGLU can deal with the problem and get better results.

In Section 3, we discussed that the instance weighting method can mitigate the noise problem in generalized labels, unfortunately it is difficult to

[4]https://github.com/BGU-CS-VIL/DeepDPM
[5]https://github.com/thuiar/CDAC-plus

estimate the noise of generalized labels accurately in the clustering algorithm. This paper uses confidence to select high confidence labeled samples and unlabeled samples. In this case, can we use confidence as weight to mitigate the noise problem? The answer is yes but we cannot get better clustering results in experiments. Taking DAC algorithm as an example, compared with the original DAC algorithm, instance weighting (regard the confidence level as the instance weight) decreases 3.31,3.79,2.63 percentage points for NMI, ARI, and ACC metrics on SNIPS dataset, and 7.19,6.59,6.16 percentage points for the three metrics on DBPedia dataset, respectively. This is because the generalized labels with low confidence level are not necessarily noise, so the instance weighting using confidence level will ignore a lot of relevant information, which makes the clustering effect appear a certain degree of degradation.

### 4.3 More evaluations

We evaluate the representation ability of DCGLU as learning better clustering features is very important for deep clustering algorithm (Caron et al., 2018). Good representation of clustering indicate the clustering results can be further improved by fine-tuning or distance-based clustering methods, etc. In the experiments, we leverage K-Means and DEC as the post clustering method based on the features learned by DCGLU, and get consistency and significant improvements on the two datasets (with p-value $< 0.01$ on paired t-test), which indicates DCGLU can learn better representations. For more details please see Appendix C.1.

The $\pi_\mathrm{p}$ in Equation 2 is the key hyper-parameter in DCGLU, we conducted a detailed analysis to guide its setting in specific applications. Fortunately, we found that using a constant $\pi_\mathrm{p}$ ($\pi_\mathrm{p} = 0.5$) can already achieve good results, which greatly reduces the conditions of DCGLU applications. For more details please see Appendix C.2.

## 5 Related Work

**Deep clustering**. Researchers have proposed many clustering methods, including center-of-mass-based clustering (MacQueen, 1967), density-based clustering (Ester et al., 1996; Comaniciu and Meer, 2002), agglomerative clustering (AG) (Gowda and Krishna, 1978) and so on, but these traditional clustering methods fix sample features, and the clustering performance of the model

tends to be poor when the extracted sample features are poor. In recent years, researchers (Xie et al., 2016; Yang et al., 2017) have combined deep neural networks to focus their work in deep clustering by feature clustering jointly. (Chang et al., 2017) proposed Deep Adaptive Cluster(DAC), which transforms the clustering problem into a binary pairwise-classification framework to determine whether samples belong to the same class. (Hadifar et al., 2019) proposed to learn discriminative features from both an autoencoder and a sentence embedding, and then update the weights of the encoder network using assignments of clustering algorithm as supervision. (Zhang et al., 2021) proposed Supporting Clustering with Contrastive Learning (SCCL) to leverage contrastive learning to promote better separation. (Ronen et al., 2022) proposed Deep Dirichlet Process Mixture(DeepDPM) to adapt to changes in the k-values of clustering categories by dynamically performing split-and-merge operations. In this paper, we focus on the influence of noise in clustering which is not explicitly investigated before.

**PU learning**. Positive-unlabeled (PU) learning has been studied for a long time (De Comité et al., 1999; Du Plessis et al., 2014, 2015; Kiryo et al., 2017). For PU learning (Liu et al., 2002) is defined as follows: given a set of positive samples $P$ and a set of unlabeled samples $U$, which contain hidden positive and negative samples, construct a binary classifier to classify the samples. PU learning is used in many natural language processing applications, (Xia et al., 2013) combined instance selection and instance weighting to apply PU learning to cross-domain sentiment classification; (Peng et al., 2019) explored the way to perform named entity recognition (NER) using only unlabeled data and named entity dictionaries; (He et al., 2020) presented a method to improve the performance of distant supervision relation extraction with PU Learning.

In this paper, inspired by PU learning, we propose a novel method (DCGLU) to improve the performance of text clustering, which 1) transform the multi-classification problem in clustering into binary classification; 2) establish a binary classification selector; 3) from the perspective of empirical risk minimization, dig the correlation between generalized labeled and unlabeled samples. Clearly, there are some major differences between PU learning and DCGLU.

# 6 Conclusion

In this paper, we propose the concept of generalized supervision and generalized labels in clustering which can help to study the impact of noise in clustering and thus improve the performance of clustering. Based on the generalized supervision and labels, we propose a deep clustering optimization method, namely Deep Clustering optimization from Generalized Labeled and Unlabeled data (DCGLU) which can be leveraged to most deep clustering methods of text. As we discussed in related works, DCGLU different from existing clustering and clustering optimization approaches. The experimental results demonstrate the necessity and effectiveness of the method.

# Acknowledgement

We would like to express gratitude to the anonymous reviewers for their kind comments. This work was supported by National Natural Science Foundation of China (No.62206308, No.61976221).

# Limitations

Due to the limitations imposed by the data representation methods, in future work, we will attempt to apply our method to image data.

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

## A Detailed Explanations

### A.1 Method of Confidence Estimation

Compared with the estimation of generalized labels, the method of obtaining confidence is relatively simple. For examples, in the method based on the cluster center, the closer the sample is to the cluster center, the higher the possibility of belonging to the center, so the confidence level can be acquired by the distance; in the method based on the similarity between samples, the higher the similarity between samples, the higher the possibility of the sample pair belonging to the same category, thus the confidence level can be acquired by the similarity between samples.

### A.2 Adapting PU Learning to Deep Clustring

According to the problem of learning from unlabeled data discussed in Section 3, we find that PU learning has more in-depth research, which can promote the identification of samples with similar characteristics of positive data among unlabeled samples. In the scenario of clustering, high-confidence labeled samples and unlabeled (low confidence) samples can be regard as the positive samples and unlabeled samples in PU learning respectively. The classical PU learning (Du Plessis et al., 2014) relies on the binary unbiased risk estimator to mine the correlation between unlabeled samples and positive samples:

$$\widehat{R}_{\mathrm{pu}}(f) = \pi_{\mathrm{p}}\widehat{R}_{\mathrm{p}}^{+}(f) - \pi_{\mathrm{p}}\widehat{R}_{\mathrm{p}}^{-}(f) + \widehat{R}_{\mathrm{u}}^{-}(f) \qquad (3)$$

where $\pi_{\mathrm{p}}$ is the prior probability of positive samples. Among Eq. (3),

$$\widehat{R}_{\mathrm{p}}^{+}(f) = \frac{1}{n_{\mathrm{p}}}\sum_{i=1}^{n_{\mathrm{p}}}\mathcal{L}\left(f\left(x_i^{\mathrm{p}}\right), +1\right) \qquad (4)$$

$$\widehat{R}_{\mathrm{p}}^{-}(f) = \frac{1}{n_{\mathrm{p}}}\sum_{i=1}^{n_{\mathrm{p}}}\mathcal{L}\left(f\left(x_i^{\mathrm{p}}\right), -1\right) \qquad (5)$$

$$\widehat{R}_{\mathrm{u}}^{-}(f) = \frac{1}{n_{\mathrm{u}}}\sum_{i=1}^{n_{\mathrm{u}}}\mathcal{L}\left(f\left(x_i^{\mathrm{u}}\right), -1\right) \qquad (6)$$

where $\{+1, -1\}$ corresponds to the labels of positive and unlabeled samples, respectively.

Here we introduce PU learning to deep clustering to help use the generalized labels and extend it to multiple classes to adapt for the application scenario of the clustering, in which high confidence labeled samples are considered as positive samples and unlabeled (low confidence) samples are considered as unlabeled samples, thus:

$$\widehat{R}_{\mathrm{p}}^{+}(f) = \frac{1}{n_{\mathrm{p}}}\sum_{i=1}^{n_{\mathrm{p}}}\mathcal{L}\left(f\left(x_i^{\mathrm{p}}\right), y_{+}^{\prime\mathrm{p}}\right) \qquad (7)$$

$$\widehat{R}_{\mathrm{p}}^{-}(f) = \frac{1}{n_{\mathrm{p}}}\sum_{i=1}^{n_{\mathrm{p}}}\mathcal{L}\left(f\left(x_i^{\mathrm{p}}\right), y_{-}^{\prime\mathrm{p}}\right) \qquad (8)$$

where, $y_{+}^{\prime\mathrm{p}}$ is the multi-class generalized label corresponding to the high confidence sample; Eq.(8) indicates that the high confidence sample does not belong to the loss of the currently assigned generalized label, in general, $y_{-}^{\prime\mathrm{p}}$ can be expressed as $y_{-}^{\prime\mathrm{p}} = 1 - y_{+}^{\prime\mathrm{p}}$. Based on the above development method, $\widehat{R}_{\mathrm{u}}^{-}(f)$ can be extended to:

$$\widehat{R}_{\mathrm{u}}^{-}(f) = \frac{1}{n_{\mathrm{u}}}\sum_{i=1}^{n_{\mathrm{u}}}\mathcal{L}\left(f\left(x_i^{\mathrm{u}}\right), y_{-}^{\prime\mathrm{u}}\right) \qquad (9)$$

However, there is no reliable multi-class generalized label for low confidence samples. Similarly, based on the maximum entropy principle, we assume that the generalized label $y^{\prime\mathrm{p}}$ of each unlabeled (low confidence) sample obeys a uniform distribution, i.e., the unlabeled sample expects the minimum under all category labels, so for the unlabeled sample, we transform Eq.(9) into:

$$\widehat{R}_{\mathrm{u}}^{-}(f) = \frac{1}{n_{\mathrm{u}}}\sum_{i=1}^{n_{\mathrm{u}}}\mathcal{L}\left(\bar{f}\left(x_i^{\mathrm{u}}\right), -1\right) \qquad (10)$$

where $\bar{f}\left(x_i^{\mathrm{u}}\right)$ is the mean of the predicted probabilities of unlabeled (low confidence) samples over all categories. The extension method, including Eq. (7), Eq.(8), and Eq.(10), can be approximated as a way to complete the learning of high confidence categorized samples and uncategorized (low confidence) samples by transforming the multi-category task split into multiple binary categories. Further, to prevent overfitting we use the non-negative risk estimator (Kiryo et al., 2017), which transforms Eq.(3) into:

$$\widehat{R}_{\mathrm{pu}}(f) = \pi_{\mathrm{p}}\widehat{R}_{\mathrm{p}}^{+}(f) + \max\left\{0, \widehat{R}_{\mathrm{u}}^{-}(f) - \pi_{\mathrm{p}}\widehat{R}_{\mathrm{p}}^{-}(f)\right\} \qquad (11)$$

By substituting Eq.(7), Eq.(8), and Eq.(10) into Eq.(11), the loss function of the optimized deep clustering model proposed in this paper can be obtained, which is called **D**eep **C**lustering optimization from **G**eneralized **L**abled and **U**nlabeled learning algorithm(**DCGLU**). It can be added as a regularization term into the currently commonly

used deep clustering algorithm, with good generalization ability:

$$\mathcal{L}_{\text{total}} = \mathcal{L}_{\text{ori}} + \lambda \cdot \widehat{R}_{\text{pu}}(f) \quad (12)$$

Where $\mathcal{L}_{\text{ori}}$ is the original loss of deep clustering algorithm, $\lambda$ is the blend factor, and $\widehat{R}_{\text{pu}}(f)$ is the loss function of deep clustering optimization model proposed in this paper.

---

**Algorithm 1** DCGLU

---

**Input**: Dataset $\mathcal{X} = \{x_i\}_{i=1}^n$, $x_i$ is the $i^{th}$ text in $\mathcal{X}$.
**Parameter**: Clustering model parameter $\theta$, hyper-parameters confidence threshold $t$ ($0 \leq t \leq 1$) and positive class-prior probability $\pi_{\text{p}}$.
**Output**: Cluster label $c_i$ for each $x_i \in \mathcal{X}$.

1: **Initialization**:Initialize the cluster network parameters;
2: **for** number of training steps **do**
3:    Sample batch $\mathcal{X}_k$ from $\mathcal{X}$, $k$ indicates the batch id;
4:    Input to clustering model $f$,Get probability of sample pairs or samples $p$;
5:    **if** $p > t$ **then**
6:       Get high confidence samples $\mathcal{X}_k^h$ from $\mathcal{X}_k$ as the positive samples $\mathcal{X}_k^p$;
7:       Calculate the empirical risk of the positive samples $\widehat{R}_{\text{p}}^+(f)$ and $\widehat{R}_{\text{p}}^-(f)$;//Eq.(7) and Eq.(8)
8:    **else**
9:       Get low confidence samples $\mathcal{X}_k^l$ from $\mathcal{X}_k$ as the unlabeled samples $\mathcal{X}_k^u$;
10:       Calculate the empirical risk of the unlabeled samples $\widehat{R}_{\text{u}}^-(f)$;//Eq.(10)
11:    **end if**
12:    **if** $\widehat{R}_{\text{u}}^-(f) - \pi_{\text{p}}\widehat{R}_{\text{p}}^-(f) \geq 0$ **then**
13:       Set gradient $\nabla_\theta \mathcal{L}_{\text{ori}} + \nabla_\theta \widehat{R}_{\text{pu}}(f)$;
14:    **else**
15:       Set gradient $\nabla_\theta \mathcal{L}_{\text{ori}} + \nabla_\theta(\pi_{\text{p}}\widehat{R}_{\text{p}}^-(f) - \widehat{R}_{\text{u}}^-(f))$;
16:    **end if**
17:    Update $\theta$ by an external SGD-like stochastic optimization algorithm;
18: **end for**
19: **for all** $\mathbf{x}_i \in \mathcal{X}$ **do**
20:    $\mathbf{l}_i := f(\mathbf{x}_i)$;
21:    $c_i := \arg\max_h(l_{ih})$;
22: **end for**

---

# B Experiment Setup

## B.1 Datasets

We conduct experiments on two publicly available text datasets. The detailed statistics are shown in Table 2.

**SNIPS** It derives from (Coucke et al., 2018) and is a personal voice assistant dataset containing 14,484 voices, divided into seven categories altogether.

**DBpedia** It contains 14 non-overlapping classes of ontology selected from DBPedia 2015 (Lehmann et al., 2015). We follow (Lin et al., 2020), which contained 1,000 samples in each classes.

## B.2 Baseline and Evaluation Metrics

We follow (Lin et al., 2020), and chose the baseline of the unsupervised part for comparison:

**KM and AG.** K-means(KM) (MacQueen, 1967) and Agglomerative Clustering(AC) (Gowda and Krishna, 1978) are classical clustering algorithms, and here we represent the sentences with the averaged pre-trained 300-dimensional word embeddings from GloVe (Pennington et al., 2014).

**SAE-KM.** We encode the sentences with the stacked autoencoder (SAE), and then execute k-means.

**BERT-KM.** We encode the sentences by BERT (Devlin et al., 2018), and then execute k-means.

**DEC.** Deep embedding clustering(DEC) (Xie et al., 2016) learns the mapping from data space to low-dimensional feature space, and uses t-distribution iteration to fine-tune clustering model.

**DCN.** Deep clustering network(DCN) (Yang et al., 2017) follows the idea of DEC and adds regularization term in the optimization process.

As an optimization strategy, in order to show the performance of our algorithm, we select two advanced deep clustering algorithms of different kinds as the main methods to test DCGLU: Deep Dirichlet Process Mixture (DeepDPM) (Ronen et al., 2022) and Deep Adaptive Clustering (DAC) (Chang et al., 2017). DeepDPM is a deep nonparametric clustering method that can adapt to k-value changes by dynamically adjusting the split and merge operations. DAC transforms the clustering problem into binary pairwise-classification framework to judge whether the samples belong to the same category.

To evaluate the experimental results, we choose three common clustering measures: Normalized Mutual Information (NMI), Adjusted Rand Index (ARI), and clustering accuracy (ACC). To calculate clustering accuracy, we use the Hungarian algorithm (Kuhn, 1955) to find the best alignment between the predicted cluster label and the ground-

| Dataset | Classes | #Training | #Validation | #Test | Vocabulary Size | Length(Avg) |
|---------|---------|-----------|-------------|-------|-----------------|-------------|
| SNIPS | 7 | 13084 | 700 | 700 | 11971 | 9.03 |
| DBPedia | 14 | 12600 | 700 | 700 | 45077 | 29.97 |

Table 2: Statistics of SNIPS and DBPedia datasets. # indicates the total number of sentences.

truth label. The higher the score of all metrics, the better the clustering performance.

### B.3 Implement Details

For both DeepDPM and DAC, we use the same BERT model to get feature vectors of text data, but as the adaptation of DeepDPM is limited for high-dimensional features, in accordance with their suggestions, we use Bert-Whitening (Su et al., 2021) to reduce the dimension of text features. Text features $H$ is reduced from 768 to 64 dimensions. For fairer comparison, we maintain the original dataset settings of (Ronen et al., 2022) and (Lin et al., 2020). Finally, we report the average results of each algorithm over ten runs.

DeepDPM is divided into clustering module and subclustering module. The model is optimized by introducing a new loss caused by Expectation-Maximization (EM) (Dempster et al., 1977) in Bayesian GMM, where every E-step is followed by a standard M-step. However, we believe that the probability of E-step generation is too incredible, and there are many misdivided samples. Therefore, we sharpen the probability of E-step generation, select high confidence labeled samples and other samples as unlabeled samples to optimize the clustering effect. DAC transforms the clustering problem into a binary pairwise-classification framework to determine whether the samples belong to the same class, so we optimize the clustering effect by treating two samples with high similarity as high-confidence labeled sample pairs and those with low similarity as unlabeled sample pairs. Our experiments conduct on pytorch[6] version 1.11.0 and 1.0.1 respectively. All experiments were performed on the NVIDIA Ge-Force RTX-2080Ti Graphical Card with 10G graphical memory.

## C Experimental Results

### C.1 Effect on clustering features

Learning better clustering features is one of the important directions of deep clustering algorithm research (Caron et al., 2018), on which the clustering effect can be further improved by fine-tuning or

---

[6]https://pytorch.org

distance-based clustering methods, etc. To explore more deeply the effect of DCGLU on clustering features, we explored the experimental results of different variants of the DAC algorithm, in which DAC-KM clusters the embedded features learned by DAC using the K-Means model, and DAC-DEC combines the idea of DEC (Xie et al., 2016) model to fine-tune and improve the clustering assignment by expectation maximization iterations.

As can be seen from Table 3 of the experimental results, each method has a certain improvement on both datasets after combining the DCGLU optimization method, which is more obvious on the DBPedia dataset, which indicates that the DCGLU method is able to optimize the clustering effect of the DAC algorithm itself, in addition to obtaining better clustering features. Note that DCGLU has limited improvement on the SNPIS dataset, and we believe this is because the performance improvement on the clustering representation cannot be efficiently transferred to the final clustering performance after post-processing, therefore the limited improvement can already indicate that DCGLU has some improvement on the clustering features.

### C.2 Effect of $\pi_p$

In the PU learning domain, $\pi_p$ denotes that the prior of positive samples can be estimated by the ratio of positive samples in the data, however, in DCGLU, the accuracy and recall of high confidence labeled sample estimates in the generalized labels of most deep clustering algorithms keep changing as the clustering algorithm learns(generally, it will be promoted first and then remain stable). This would result in $\pi_p$ changing dynamically throughout the training process and thus difficult to estimate. Fortunately, we found that using a constant $\pi_p$ can already achieve good results, which greatly reduces the conditions of DCGLU applications.

The effects of different $\pi_p$ on the experimental results are shown in Figure 2, where $\pi_p \in \{0.1, 0.3, 0.5, 0.7, 0.9\}$. It can be seen that for different $\pi_p$, the three evaluation metrics of the two data sets show the same trend of change, and the optimal effect is reached at $\pi_p = 0.5$. When $\pi_p$

| Dataset | SNIPS | | | DBPedia | | |
|---|---|---|---|---|---|---|
| Method | NMI | ARI | ACC | NMI | ARI | ACC |
| DAC | $75.51 \pm 4.93$ | $67.13 \pm 7.10$ | $77.17 \pm 6.61$ | $72.73 \pm 3.57$ | $53.01 \pm 5.04$ | $63.27 \pm 5.74$ |
| DAC* | $\mathbf{79.92} \pm 4.40$ | $\mathbf{70.67} \pm 6.89$ | $\mathbf{79.64} \pm 6.22$ | $\mathbf{74.25} \pm 2.57$ | $\mathbf{56.93} \pm 4.36$ | $\mathbf{66.90} \pm 4.96$ |
| DAC-KM | $86.65 \pm 3.35$ | $83.14 \pm 5.34$ | $91.49 \pm 3.51$ | $84.93 \pm 2.45$ | $73.74 \pm 4.94$ | $80.27 \pm 5.46$ |
| DAC-KM* | $\mathbf{87.60} \pm 1.64$ | $\mathbf{84.70} \pm 2.06$ | $\mathbf{92.52} \pm 1.18$ | $\mathbf{85.26} \pm 2.53$ | $\mathbf{74.34} \pm 5.34$ | $\mathbf{82.37} \pm 5.95$ |
| DAC-DEC | $87.17 \pm 2.87$ | $83.79 \pm 4.46$ | $91.78 \pm 3.02$ | $85.59 \pm 2.42$ | $74.74 \pm 4.68$ | $81.18 \pm 5.31$ |
| DAC-DEC* | $\mathbf{87.52} \pm 1.49$ | $\mathbf{84.59} \pm 1.73$ | $\mathbf{92.44} \pm 1.00$ | $\mathbf{86.52} \pm 2.69$ | $\mathbf{76.69} \pm 5.23$ | $\mathbf{83.14} \pm 5.77$ |

Table 3: The clustering results for variants of DAC.

| Method-Dataset | DCGLU Time | Base Time | Proportion(DCGLU Time/Base Time) |
|---|---|---|---|
| DAC-Snips | 10.74s | 829.22s | 1.29 |
| DAC-DBPedia | 10.30s | 1193.92s | 0.86 |
| DeepDPM-Snips | 104.60s | 1832.19s | 5.70 |
| DeepDPM-DBPedia | 102.29s | 2855.89s | 3.58 |

Table 4: Experimental results of efficiency cost of DCGLU

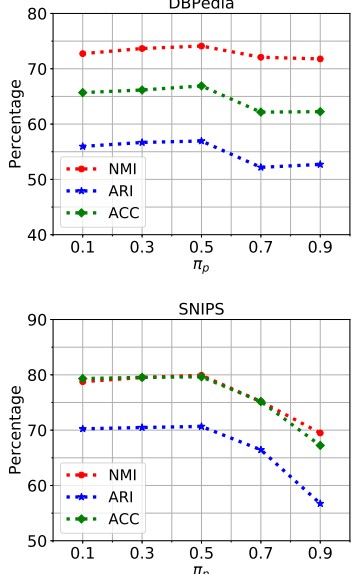

Figure 2: Experimental results of optimization of DAC algorithm given $\pi_{\mathrm{p}} \in \{0.1, 0.3, 0.5, 0.7, 0.9\}$.

We conduct a runtime complexity experiment, and Table 4 shows the run time. "Proportion(DCGLU Time/Base Time)" indicates the proportion of the calculation time of DCGLU in the entire program running time. As can be seen from the table, our method does not bring much additional time over the entire run time. This is because DCGLU has a time complexity of O(N), and at the same time, the additional spatial complexity brought by DCGLU is O(N).

is overestimated, there is a significant decrease in the experimental results, while the experimental results tend to be stable when $\pi_{\mathrm{p}}$ is small, so we suggest not to use a larger $\pi_{\mathrm{p}}$. This is because in the clustering process, there are still a small number of errors in the high confidence labeled samples. Hence, under the condition of ensuring performance, a smaller $\pi_{\mathrm{p}}$ should be used to reduce the probability of introducing errors in DCGLU, so as to improve model stability and clustering effect.

## C.3 Effect of time efficiency

DCGLU does not increase the complexity of the primary clustering algorithm in terms of efficiency.