# OpenReview forum: "Improved Training of Deep Text Clustering"
_EMNLP/2023/Conference — EMNLP 2023 Findings_

### Official Review · Reviewer_ynRi · 2023-07-19

**Soundness:** 4

**Excitement:**

3: Ambivalent: It has merits (e.g., it reports state-of-the-art results, the idea is nice), but there are key weaknesses (e.g., it describes incremental work), and it can significantly benefit from another round of revision. However, I won't object to accepting it if my co-reviewers champion it.

**Paper Topic And Main Contributions:**

This paper is about deep text clustering, which is the task of grouping text data into different categories without using any labels. The paper addresses the problem of noise in the generalized labels that are used to provide weak supervision for deep clustering models. Generalized labels are derived from different sources such as cluster centers, similarity matrices, or clustering assignments, and they may contain errors due to the limitation of clustering accuracy. The paper proposes a novel method called Deep Clustering optimization from Generalized Labeled and Unlabeled data (DCGLU), which reduces the impact of noise in generalized labels by dividing the samples into high confidence and low confidence ones, and applying a binary classification framework inspired by positive and unlabeled learning1. The paper makes the following contributions:
1. It introduces the concept of generalized supervision and generalized labels in clustering, which can help to study the impact of noise in clustering and thus improve the performance of deep clustering.
2. It proposes a general deep clustering optimization method, which can be leveraged to reduce the impact of noise in clustering and can be applied to most deep clustering methods of text.
3. It conducts experiments on two text datasets using two strong deep clustering methods, and shows that the proposed method can improve the clustering features and clustering results of the baselines.

**Reasons To Accept:**

1. It tackles an important and challenging problem of deep text clustering, which is a useful technique for many NLP applications such as information retrieval, text summarization, and topic modeling.
2. It proposes a novel and general method that can be easily integrated with existing deep clustering methods, and does not require any additional parameters or hyperparameters.
3. It provides a clear and rigorous theoretical analysis of the proposed method, and shows its connection to the empirical risk minimization principle.
4. It conducts extensive experiments on two text datasets using two strong deep clustering methods, and demonstrates the effectiveness and robustness of the proposed method1.
5. It compares the proposed method with several baselines and variants, and analyzes the impact of different factors such as confidence level, prior probability, and feature dimensionality.

**Reasons To Reject:**

1. It does not provide a clear definition or formalization of the concept of generalized supervision and generalized labels, which may cause confusion or misunderstanding for the readers.
2. It does not compare the proposed method with other existing methods that also aim to reduce the noise in clustering, such as instance weighting or noise filtering, which may limit the comprehensiveness and fairness of the evaluation.
3. It does not conduct ablation studies or sensitivity analysis to show the contribution of each component or parameter of the proposed method, which may affect the interpretability and reproducibility of the results.
4. It does not discuss the limitations or potential drawbacks of the proposed method, such as the scalability or efficiency issues, or the possible failure cases or scenarios.
5. It may not provide enough novelty or significance for the NLP community, as deep text clustering is a well-studied problem and the proposed method is based on existing techniques such as PU learning and binary classification.
6. It may not provide enough clarity or rigor for the NLP community, as some of the concepts and methods are not well-defined or justified, and some of the experiments and analyses are not well-designed or reported.

Over all, a finding paper may be suitable for this paper.

**Reproducibility:**

3: Could reproduce the results with some difficulty. The settings of parameters are underspecified or subjectively determined; the training/evaluation data are not widely available.

**Reviewer Confidence:**

4: Quite sure. I tried to check the important points carefully. It's unlikely, though conceivable, that I missed something that should affect my ratings.

---

> ### Author Rebuttal · Authors · 2023-08-29
>
> We sincerely thank you for you constructive comments and generous supports! For your questions, we have the following replies:
>
> > Q1. It does not provide a clear definition or formalization of the concept of generalized supervision and generalized labels, which may cause confusion or misunderstanding for the readers.
> >
>
> A1. In essence, our method performs text clustering in unsupervised scenarios, but we find that many clustering methods use high-confidence samples for fitting training in disguised form, and this high-confidence sample can be understood as a label in supervised learning. Therefore, we find that there exists a phenomenon of using generalized labels for generalized supervised learning in clustering algorithms. We will update the statements in the paper and redefine the concepts of generalized supervision and generalized labeling.
>
> > Q2. It does not compare the proposed method with other existing methods that also aim to reduce the noise in clustering, such as instance weighting or noise filtering, which may limit the comprehensiveness and fairness of the evaluation.
> >
>
> A2. We show the influence of instance weighting on the DAC method in lines 217-229 of this paper. For your convenience, we summarize the results in the table below.
>
> | Dataset |  | SNIPS |  |  | DBPedia |  |
> | :---: | :---: | :---: | :---: | :---:| :---: | :---: |
> | Method | NMI | ARI | ACC | NMI | ARI | ACC |
> | DAC | 75.51 ± 4.93 | 67.13 ± 7.10 | 77.17 ± 6.61 | 72.73 ± 3.57 | 53.01 ± 5.04 | 63.27 ± 5.74 |
> | DAC + Instance Weighting | 72.2 ± 6.03 | 63.34 ± 9.02 | 74.54 ± 7.94 | 65.54 ± 3.61 | 46.42 ± 4.26 | 57.11 ± 4.53 |
> | DAC + DCGLU | 79.92 ± 4.40 | 70.67 ± 6.89 | 79.64 ± 6.22 | 74.25 ± 2.57 | 56.93 ± 4.36 | 66.90 ± 4.96 |
>
> > Q3. It does not conduct ablation studies or sensitivity analysis to show the contribution of each component or parameter of the proposed method, which may affect the interpretability and reproducibility of the results.
> >
>
> A3. In the appendix, we show the influence of the ablation experiment and explore the influence of the setting of $\pi_p$ on the experiment. For the definition parameters of confidence, DAC native method automatically adjusts the definition of high confidence samples, which we directly followed, while DeepDPM method requires us to manually set confidence threshold. In future work, we will explore the induction study of automatic confidence threshold.
>
> > Q4. It does not discuss the limitations or potential drawbacks of the proposed method, such as the scalability or efficiency issues, or the possible failure cases or scenarios.
> >
>
> A4. Thank you for your valuable concern. We would like to answer your question as follows.
>
> - **scalability or efficiency issues:** Our method does not increase the complexity of the primary clustering algorithm in terms of efficiency. We did a runtime complexity experiment, and the following table shows the run time. “Proportion(DCGLU Time/Base Time)” indicates the proportion of the calculation time of DCGLU in the entire program running time. As can be seen from the table, our  method does not bring much additional time over the entire run time. This is because DCGLU has a time complexity of O(N), and at the same time, the additional spatial complexity brought by DCGLU is O(N). We will release our code.
>
>
>     | Method-Dataset | DCGLU Time | Base Time | Proportion(DCGLU Time/Base Time) |
>     | :---: | :---: | :---: | :---: |
>     | DAC-Snips | 10.74s | 829.22s | 1.29% |
>     | DAC-DBPedia | 10.30s | 1193.92s | 0.86% |
>     | DeepDPM-Snips | 104.60s | 1832.19s | 5.70% |
>     | DeepDPM-DBPedia | 102.29s | 2855.89s | 3.58% |
> - **possible failure cases or scenarios**: For example, if the high-confidence sample is not trusted, the high-confidence sample can not bring substantial generalized supervised feedback, resulting in the failure of our optimization algorithm, but in fact, this problem exists for all clustering algorithms.
>
> > Q5. It may not provide enough novelty or significance for the NLP community, as deep text clustering is a well-studied problem and the proposed method is based on existing techniques such as PU learning and binary classification.
> >
>
> A5. Although we integrate existing mature techniques, the fact that we have identified a problem prevalent in deep clustering, generalised and abstracted it, and described it formally in a mathematical language is already a clear contribution in itself. The reason for utilising PU learning is because we have found it to be a good solution to the above problem and have experimentally demonstrated the improvement in clustering performance by solving the problem.
>
> > Q6. It may not provide enough clarity or rigor for the NLP community, as some of the concepts and methods are not well-defined or justified, and some of the experiments and analyses are not well-designed or reported.
> >
>
> A6. We will proofread the paper carefully. Redefine and rationalise the concepts and methods we present. And add additional experiments and analyses to the new version of the paper.

---

### Official Review · Reviewer_kABq · 2023-07-25

**Soundness:** 4

**Excitement:**

3: Ambivalent: It has merits (e.g., it reports state-of-the-art results, the idea is nice), but there are key weaknesses (e.g., it describes incremental work), and it can significantly benefit from another round of revision. However, I won't object to accepting it if my co-reviewers champion it.

**Missing References:**

There are hardly any references to deep clustering in the text domain. The following publications would be interesting:

[1] Supporting Clustering with Contrastive Learning (Zhang et.al.)
[2] A Self-Training Approach for Short Text Clustering (Hadifar et.al)

The current related work focuses on deep clustering for images, which would be fine if the paper would not be submitted with the title 'deep text clustering'.

**Paper Topic And Main Contributions:**

The authors enhance deep clustering algorithms by integrating Positive and Unlabeled Learning. They showcase, that this minor adaption to the loss function improves existing deep clustering models.

**Questions For The Authors:**

1) Did you evaluate your algorithm in other domains (e.g. computer vision)? If not, why did you decide only working with text data?

2) Did you perform runtime complexity experiments? It would be vital to showcase, that the extension does not suffer from high computational costs.

3) In your method section you mentioned 'instance weighting' (127-129) as a comparable approach, yet you did not evaluate your approach against this technique. Does you method perform better than the aforementioned algorithm?

**Reasons To Accept:**

The simple extension to deep clustering models has demonstrated superior performance across all evaluated datasets, making it applicable to various scenarios and improving overall results. The paper is well-motivated, written with clarity, and easy to understand.

**Reasons To Reject:**

The novelty of the approach is somewhat limited, as it is a combination of two already existing and well-established techniques.
The proposed model uses Learning from Positive and Unlabeled Data and evaluates it on a deep clustering framework.
Furthermore, the labeling of the method as 'Deep Text Clustering' seems unclear, as it is capable of performing well in other domains too. Apart from the BERT model as the backbone, which could be substituted with any pre-trained model (e.g., ResNet), no component is specifically tailored to the NLP domain.

**Reproducibility:**

4: Could mostly reproduce the results, but there may be some variation because of sample variance or minor variations in their interpretation of the protocol or method.

**Reviewer Confidence:**

4: Quite sure. I tried to check the important points carefully. It's unlikely, though conceivable, that I missed something that should affect my ratings.

**Typos Grammar Style And Presentation Improvements:**

While the paper is well-written, it would be beneficial to make the main paper more self-contained by moving vital parts of the algorithm, such as equation (12), to the main body instead of placing them in the Appendix. This would enhance the overall clarity and accessibility of the paper.

---

> ### Author Rebuttal · Authors · 2023-08-29
>
> We sincerely thank you for you constructive comments and generous supports! For your questions, we have the following replies:
>
> > Q1. Did you evaluate your algorithm in other domains (e.g. computer vision)? If not, why did you decide only working with text data?
> >
>
> A1. We have conducted experiments on image datasets and found that the performance improvement is limited (not significant), but it has shown good results on text datasets. We leave the analysis of the insignificant improvement on images as future work.
>
> > Q2. Did you perform runtime complexity experiments? It would be vital to showcase, that the extension does not suffer from high computational costs.
> >
>
> A2. We did a runtime complexity experiment, and the following table shows the run time. “Proportion(DCGLU Time/Base Time)” indicates the proportion of the calculation time of DCGLU in the entire program running time. As can be seen from the table, our  method does not bring much additional time over the entire run time. This is because DCGLU has a time complexity of O(N), and at the same time, the additional spatial complexity brought by DCGLU is O(N). We will release our code.
>
> | Method-Dataset | DCGLU Time | Base Time | Proportion(DCGLU Time/Base Time) |
> | :---: | :---: | :---: | :---: |
> | DAC-Snips | 10.74s | 829.22s | 1.29% |
> | DAC-DBPedia | 10.30s | 1193.92s | 0.86% |
> | DeepDPM-Snips | 104.60s | 1832.19s | 5.70% |
> | DeepDPM-DBPedia | 102.29s | 2855.89s | 3.58% |
>
> > Q3. In your method section you mentioned 'instance weighting' (127-129) as a comparable approach, yet you did not evaluate your approach against this technique. Does you method perform better than the aforementioned algorithm?
> >
>
> A3. We show the influence of instance weighting on the DAC method in lines 217-229 of this paper. For your convenience, we summarize the results in the table below.
>
> | Dataset |  | SNIPS |  |  | DBPedia |  |
> | :---: | :---: | :---: | :---: | :---: | :---: | :---: |
> | Method | NMI | ARI | ACC | NMI | ARI | ACC |
> | DAC | 75.51 ± 4.93 | 67.13 ± 7.10 | 77.17 ± 6.61 | 72.73 ± 3.57 | 53.01 ± 5.04 | 63.27 ± 5.74 |
> | DAC + Instance Weighting | 72.2 ± 6.03 | 63.34 ± 9.02 | 74.54 ± 7.94 | 65.54 ± 3.61 | 46.42 ± 4.26 | 57.11 ± 4.53 |
> | DAC + DCGLU | 79.92 ± 4.40 | 70.67 ± 6.89 | 79.64 ± 6.22 | 74.25 ± 2.57 | 56.93 ± 4.36 | 66.90 ± 4.96 |
>
> > Q4. It would be beneficial to make the main paper more self-contained by moving vital parts of the algorithm, such as equation (12), to the main body instead of placing them in the Appendix. This would enhance the overall clarity and accessibility of the paper. There are hardly any references to deep clustering in the text domain.
> >
>
> A4. Thank you for your kind correction. We put formula 12 in the appendix because it is relatively less important. We will revise the expressions and citations in the paper as we will have another page to revise the paper after acceptence . Thank you very much for your comments.

---

### Official Review · Reviewer_xqXP · 2023-08-06

**Soundness:** 2

**Excitement:**

2: Mediocre: This paper makes marginal contributions (vs non-contemporaneous work), so I would rather not see it in the conference.

**Paper Topic And Main Contributions:**

The authors propose the concept of generalized supervision with generalized label, and introduce a general deep clustering optimization method, which can be leveraged to reduce the impact of noise in clustering. The experiments show that the proposed DCGLU can improve the clustering performance over other baselines.

**Questions For The Authors:**

1.	In lines 139 to 142, what does the ‘contrary’ mean? Is it for low confidence samples?

**Reasons To Accept:**

1.	The proposed method can be added to other methods as a strategy to improve the clustering performance.
2.	The experimental results show the proposed DCGLU can improve the clustering features and clustering results.


**Reasons To Reject:**

1.	The specific problems in the existing methods have not been fully discussed.
2.	The presentation of the article is not very clear.
3.	The experiments lack of ablation study.

**Reproducibility:**

3: Could reproduce the results with some difficulty. The settings of parameters are underspecified or subjectively determined; the training/evaluation data are not widely available.

**Reviewer Confidence:**

3: Pretty sure, but there's a chance I missed something. Although I have a good feel for this area in general, I did not carefully check the paper's details, e.g., the math, experimental design, or novelty.

---

> ### Author Rebuttal · Authors · 2023-08-29
>
> We sincerely thank you for you constructive comments and generous supports! For your questions, we have the following replies:
>
> > Q1. In lines 139 to 142, what does the 'contrary' mean? Is it for low confidence samples?
> >
>
> A1. The word "contrary" means "conversely" in our paper. In general, we cannot prediction the labels of the samples other than high-confidence samples as the current model is confused about them. But this does not mean that these samples are all negative samples, as there may be positive samples inside (just because the current model does not have the ability to correctly recognize them). So we consider it as an unlabeled sample.
>
> > Q2. The specific problems in the existing methods have not been fully discussed.
> >
>
> A2. Our paper discusses two classical deep clustering algorithms, GMM and spectral clustering. We found that many clustering algorithms contain generalized supervised information, such as clustering center, distance between samples, etc. We believe that the strategy of directly fitting high and low confidence samples in clustering algorithms is not reliable, so we combined PU learning to process low confidence samples as unlabeled samples.
>
> > Q3. The presentation of the article is not very clear.
> >
>
> A3. About unclear representation, we are not sure which part or aspect you are referring to. We will carefully proofread the paper. We guess you don't understand the definition of generalized supervision or generalized label. In fact, in essence, our work defines the way to fit the cluster center in deep clustering as generalized supervision, and the information such as cluster center and sample similarity is defined as generalized label. Then the samples with high confidence are taken as positive samples and the samples with low confidence are taken as unlabeled samples to optimize the clustering performance and text features. That's the core idea of what we do. Finally, Thank you for your kind correction. We will revise the expressions in the paper according to your comments. Thank you very much for your comments.
>
> > Q4. The experiments lack of ablation study.
> >
>
> A4. In the appendix C, we show the influence of the ablation experiment and explore the influence of the setting of $\pi_p$ on the experiment.

---

### Meta-Review · Area_Chair_pmvq · 2023-09-15

**Recommendation:** 1

**Metareview:**

This paper delves into the problem of optimizing deep clustering from an empirical risk minimization perspective, with a focus on utilizing the correlation between samples.

The reviewers have expressed varying opinions on this paper. After a thorough examination of the manuscript, I have identified several concerns that need to be addressed. These include limited novelty, poorly presented details (e.g., Deep Text Clustering, generalized supervision and generalized labels), the absence of SOTA baseline comparisons, and a lack of ablation study. Additionally, while the authors acknowledge some prior shortcomings in the Introduction, these issues should be more clearly elucidated and appropriately cited.

To address these concerns and enhance the quality, I recommend that the authors thoroughly revise their work and resubmit it for further evaluation.

---

### Decision · Program_Chairs · 2023-10-07

**Decision:**

Accept-Findings

**Comment:**

This paper delves into the problem of optimizing deep clustering from an empirical risk minimization perspective, with a focus on utilizing the correlation between samples.

The reviewers have expressed varying opinions on this paper. After a thorough examination of the manuscript, I have identified several concerns that need to be addressed. These include limited novelty, poorly presented details (e.g., Deep Text Clustering, generalized supervision and generalized labels), the absence of SOTA baseline comparisons, and a lack of ablation study. Additionally, while the authors acknowledge some prior shortcomings in the Introduction, these issues should be more clearly elucidated and appropriately cited.

To address these concerns and enhance the quality, I recommend that the authors thoroughly revise their work and resubmit it for further evaluation.